# Efficacy and Safety of Ticagrelor versus Clopidogrel in Dialysis Patients with Coronary Syndromes: A Systematic Review and Meta-Analysis

**DOI:** 10.3390/jcm12155011

**Published:** 2023-07-30

**Authors:** Alexandru Burlacu, Mariana Floria, Crischentian Brinza, Adrian Covic

**Affiliations:** 1Institute of Cardiovascular Diseases “Prof. Dr. George I.M. Georgescu”, 700503 Iasi, Romania; alexandru.burlacu@umfiasi.ro; 2Faculty of Medicine, University of Medicine and Pharmacy “Grigore T Popa”, 700115 Iasi, Romania; floria.mariana@umfiasi.ro (M.F.); adrian.covic@umfiasi.ro (A.C.); 3Medical Clinic, “Sf. Spiridon” Emergency Hospital, 700111 Iasi, Romania; 4Nephrology Clinic, Dialysis and Renal Transplant Center, “C.I. Parhon” University Hospital, 700503 Iasi, Romania

**Keywords:** ticagrelor, clopidogrel, dialysis, end-stage kidney disease, adverse outcomes, bleedings

## Abstract

(1) Background: The optimal antiplatelet therapy for end-stage kidney disease (ESKD) patients on chronic dialysis presenting with acute or chronic coronary syndromes (ACS or CCS) remains uncertain. This meta-analysis aimed to compare the efficacy and safety endpoints of ticagrelor and clopidogrel in ESKD patients requiring dialysis and presenting with ACS or CCS. (2) Methods: Studies were included comparing ticagrelor and clopidogrel in ESKD patients on chronic dialysis with ACS or CCS. The primary composite efficacy outcome was a combination of all-cause and cardiovascular mortality, recurrent myocardial infarction or coronary revascularization, and ischemic or hemorrhagic stroke. The primary safety outcome was major and non-major bleeding events. (3) Results: Five observational studies met the eligibility criteria. The pooled analysis showed no significant difference in the primary composite efficacy outcome between ticagrelor and clopidogrel (*p* = 0.40). Similarly, the 2 groups had no significant differences in all-cause mortality (*p* = 0.82) or cardiovascular mortality (*p* = 0.79). Ticagrelor did not show a significantly different risk of coronary revascularization (*p* = 0.35) or recurrent myocardial infarction (*p* = 0.41) compared to clopidogrel. Also, the risk of stroke was similar (*p* = 0.21). The 2 groups had no significant difference in the primary composite safety outcome (*p* = 0.22) or major bleeding events (*p* = 0.27). (4) Conclusions: In ESKD patients on chronic dialysis with ACS or CCS, there was no significant difference in efficacy or safety outcomes between ticagrelor and clopidogrel. Further randomized controlled trials are needed to establish the optimal antiplatelet therapy in this population.

## 1. Introduction

The optimal antiplatelet therapy for patients with chronic kidney disease (CKD) undergoing percutaneous coronary intervention (PCI) for both acute and chronic coronary syndromes (ACS and CCS, respectively) has been extensively investigated in clinical trials [1]. However, there are limited data on the optimal P2Y12 receptor inhibitors, particularly for more potent drugs such as ticagrelor or prasugrel, in patients with end-stage kidney disease (ESKD) receiving chronic dialysis [1,2].

The recommendations of the European Society of Cardiology (ESC) guidelines concerning the optimal choice of P2Y12 receptor inhibitors in patients requiring chronic dialysis are equivocal [3,4].

On the one hand, the ESC guidelines for patients with ST-elevation myocardial infarction advise against the use of ticagrelor in individuals with an estimated glomerular filtration rate (eGFR) less than 15 mL/min/1.73 m^2^ [3]. On the other hand, the latest guidelines for ACS without ST elevation recommend no dose adjustment for ticagrelor in patients with CKD (regardless of renal impairment severity) [4]. 

The absence of well-designed randomized controlled clinical trials or large-scale studies examining the safety and efficacy of ticagrelor and prasugrel in chronic dialysis constitutes a significant research gap. The lack of such studies makes it difficult to determine the optimal antiplatelet therapy for this patient population, potentially leading to worse treatment outcomes and adverse events. Further research is needed to determine the safety and efficacy of existing antiplatelet therapies in dialysis patients; efforts should be made to facilitate and promote such studies [3,4]. 

Clinical studies have demonstrated a greater risk of adverse cardiovascular events in patients with CKD who present with ACS, particularly ESKD patients. Patients with advanced CKD stages have a 60% increased long-term risk of cardiovascular death, myocardial infarction, or stroke compared to those with normal renal function or mild CKD. Furthermore, the risk of major bleeding was three times higher in patients with severe CKD than those without or with mild CKD [5]. In-hospital major adverse cardiovascular events (MACE) were reported in 32.5% of patients with an estimated glomerular filtration rate (eGFR) less than 30 mL/min/1.73 m^2^ (including dialysis) who underwent PCI for ACS [6]. These findings underscore the need for tailored therapy and antithrombotic regimen in patients with ESKD to reduce the ischemic risk following PCI while minimizing the risk of bleeding [2].

Although ESKD is a risk factor for adverse events in PCI and ACS or CCS, these patients were systematically excluded from major trials comparing the efficacy and safety of different antiplatelet regimens [2]. The Ticagrelor versus Clopidogrel in Patients with Acute Coronary Syndromes (PLATO) trial reported a significantly lower incidence of the composite outcome (death from vascular causes, myocardial infarction, or stroke) in the ticagrelor arm compared to clopidogrel (9.8% and 11.7%, respectively, *p* < 0.001). However, patients requiring dialysis were excluded from the trial [7,8]. In the Ticagrelor With Aspirin or Alone in High-Risk Patients After Coronary Intervention (TWILIGHT) study, ticagrelor alone after 3 months of DAPT in high-risk PCI patients was associated with a lower risk of bleeding events (HR 0.49%, 95% CI, 0.33–0.74), with similar efficacy compared to the aspirin plus ticagrelor arm. Although high-risk patients for bleeding and ischemic events were enrolled in the study, CKD patients on chronic dialysis were excluded from the analysis [9,10].

Regarding patients with ACS and eGFR ≤ 30 mL/min/1.73 m^2^ who underwent PCI, some authors reported no significant difference in the incidence of the primary composite outcome (cardiovascular death, recurrent myocardial infarction, and non-fatal ischemic stroke) between ticagrelor and clopidogrel arms (HR 0.78, 95% CI, 0.46–1.33, *p* = 0.367) [11]. However, patients receiving ticagrelor had a higher risk of bleeding than those receiving clopidogrel (HR 3.01, 95% CI, 1.81–5.62, *p* = 0.01). Although the effect was consistent irrespective of dialysis requirement, it should be noted that only 18.1% (n = 50) of patients were on chronic dialysis. Thus, extrapolating these results to all ESKD patients may not be appropriate [11].

Consequently, we aimed to systematically review the literature on the efficacy and safety of ticagrelor therapy compared to clopidogrel in ESKD patients on chronic dialysis, requiring antiplatelet therapy in the context of ACS or CCS.

## 2. Materials and Methods

In the present systematic review, we used the updated Preferred Reporting Items for Systematic Reviews and Meta-Analyses (PRISMA) guidelines (including for the search process and data collection and reporting) [12]. The protocol of the systematic review and meta-analysis was registered in the PROSPERO database (CRD42023422545).

### 2.1. Data Sources and Search Strategy

We conducted a search to find studies relevant to our research question in MEDLINE (PubMed), Embase, Scopus, and Cochrane databases between 5 February 2023 and 10 April 2023. The search did not involve any language filters. Apart from the above sources, we also searched for additional citations in Google Scholar and ClinicalTrials.gov databases. Furthermore, we examined the references of representative studies to identify studies that met our eligibility criteria. To create a comprehensive search strategy, we utilized various combinations of keywords and controlled vocabulary (MeSH terms for MEDLINE and Emtree for Embase), as follows: “ticagrelor”, “clopidogrel”, “antiplatelet therapy”, “dual antiplatelet therapy”, “platelet inhibitor”, “antiaggregant”, “chronic kidney disease”, “end-stage kidney disease”, “renal impairment”, “dialysis”, “hemodialysis”, “peritoneal dialysis”, “adverse outcomes”, “adverse events”, “major adverse cardiovascular events”, “mortality”, “death”, “coronary artery revascularization”, “restenosis”, “myocardial infarction”, “stroke”, “target lesion revascularization”, “hemorrhage”, “bleeding”, “major bleeding”, and “non-major bleeding”. The search strategy in all databases was presented in Appendix A.

### 2.2. Eligibility Criteria and Outcomes

Before conducting the search and data extraction, we established specific inclusion and exclusion criteria for the present systematic review. Two independent investigators used these criteria to decide which retrieved studies were eligible and were included in the analysis. The following inclusion criteria were applied: (1) randomized controlled studies and observational studies; (2) adult patients ≥ 18 years were enrolled; (3) patients with ESKD requiring renal replacement therapy (hemodialysis or peritoneal dialysis) were analyzed; (4) studies which enrolled patients with ACS or CCS; (5) studies compared efficacy and/or safety endpoints of ticagrelor and clopidogrel. Also, some exclusion criteria were pre-defined: case reports and case series, editorials, overlapping populations, unpublished data, meta-analyses, and missing outcome data.

The primary composite efficacy endpoint consisted of all-cause and cardiovascular mortality, recurrent myocardial infarction or coronary revascularization, and ischemic or hemorrhagic stroke. The primary safety outcome was a composite of major and non-major bleeding events. Secondary efficacy and safety outcomes comprised the individual components of the primary composite outcomes, as well as gastrointestinal and intracerebral hemorrhages.

### 2.3. Data Collection and Synthesis

Two independent investigators extracted the following data from included studies after eligibility assessment: first author and publication year, design of the study, number of patients included in ticagrelor and clopidogrel arms, clinical setting (ACS, CCS or both), comorbidities of enrolled patients, investigated outcomes (including the number of events in each arm), and follow-up duration.

The Review Manager (RevMan) version 5.4.1 (Nordic Cochrane Centre, The Cochrane Collaboration, 2020, Copenhagen, Denmark) was used to obtain the pooled effect size, the odds ratio (OR), and corresponding 95% confidence intervals (CIs). In the case of dichotomous data, the random-effect model and Mantel–Haenszel method were applied. The level of heterogeneity among the included studies was assessed using the I^2^ statistics, which was categorized as low (0–25%), moderate (26–50%), high (51–75%), and very high (>75%). A *p*-value below 0.05 was considered statistically significant.

We performed sensitivity analysis by sequentially excluding studies that reported specific outcomes (e.g., all-cause or cardiovascular mortality, specific sites of bleeding events) and in patients presenting exclusively with ACS, including acute myocardial infarction.

### 2.4. Quality Assessment

In the present systematic review, we utilized the Newcastle–Ottawa Scale (NOS) to assess the quality of nonrandomized studies [13]. NOS is a rating system based on stars that evaluate studies based on three key domains: selection, comparability of groups, and investigated outcomes. Each domain includes a set of essential questions, and stars are awarded based on the overall quality judgment [13].

## 3. Results

A systematic search was conducted across the specified databases, yielding an initial pool of 935 records. Afterward, duplicate publications were excluded, resulting in a refined dataset consisting of 464 references. These articles were assessed based on predefined inclusion and exclusion criteria. Two independent investigators then conducted a thorough screening of titles and abstracts, followed by an evaluation of the full-text articles. Following the full-text screening, five studies met the eligibility criteria and were included in the present analysis, as was presented in the search flowchart (Figure 1).

Characteristics of analyzed studies, including publication year, study design, clinical setting of enrolled patients, and their age, as well as follow-up period, were summarized in Table 1. Reported results and investigated outcomes are described in Table 2. All studies had an observational design [14,15,16,17,18]. Three out of five studies compared the efficacy and safety of ticagrelor and clopidogrel exclusively in patients presenting with ACS, including acute myocardial infarction [15,16,18]. The follow-up duration in reported studies was primarily 12 months or until outcomes occurrence [15,16,17]. One study has also analyzed in-hospital outcomes of patients with ST-elevation myocardial infarction (STEMI) and non-ST-elevation myocardial infarction (NSTEMI) [15]. 

In the pooled analysis, the primary composite efficacy endpoint occurred in 52.7% (n = 1167) of patients receiving ticagrelor, compared to 73.3% (n = 39371) from the clopidogrel group. Although the number of adverse events tended to be lower in the ticagrelor arm, it was not statistically significant, OR = 0.84, 95% CI, 0.57–1.25, *p* = 0.40 (Figure 2A). However, the heterogeneity across analyzed studies was very high (I^2^ = 93%). In the sensitivity analysis, we excluded the study by Jain et al. [14], thus reducing the heterogeneity (I^2^ = 67%), but the overall effect maintained nonsignificant (*p* = 0.83).

Concerning secondary efficacy endpoints, all-cause mortality was reported in 20.8% of patients from the ticagrelor group, compared to 30.1% in the case of patients receiving clopidogrel. Nevertheless, the effect did not reach statistical significance, OR = 0.95, 95% CI, 0.60–1.49, *p* = 0.82 (Figure 2B), but with very high heterogeneity, I^2^ = 93%. Although the exclusion of the study by Jain et al. [14] resulted in a reduction in heterogeneity to a low level (I^2^ = 18%), all-cause mortality was similar in both treatment groups (*p* = 0.19). Also, cardiovascular mortality tended to be lower in patients receiving ticagrelor than in the clopidogrel group (11.0% vs. 13.5%). However, the overall effect was also statistically nonsignificant, OR = 0.94%, 95% CI, 0.60–1.47, *p* = 0.79 (Figure 2C). Removing the study Jain et al. [14] from the analysis, as a source of heterogeneity, did not impact endpoint occurrence.

The risk of coronary revascularization was similar in both treatment groups OR = 0.94, 95% CI, 0.83–1.07, *p* = 0.35 (Figure 2D). However, data on coronary revascularization were reported in only two studies, and the results should be cautiously interpreted and extrapolated. Data on recurrent myocardial infarction were available in four studies. Ticagrelor therapy was associated with a similar risk of recurrent myocardial infarction as clopidogrel, OR 0.93, 95% CI, 0.77–1.11, *p* = 0.41 (Figure 2E), with low heterogeneity (I^2^ = 0%).

Stroke occurred in 8.5% of patients receiving ticagrelor, compared to 15.6% of patients from the clopidogrel arm. Even though the stroke rate was lower in the ticagrelor group, it was not significant, OR = 0.70, 95% CI, 0.41–1.22, *p* = 0.21 (Figure 2F), with high heterogeneity (I^2^ = 56%). We sequentially removed studies from the analysis and identified the study by Tung et al. [18] as a source of heterogeneity (the effect maintained nonsignificant after the exclusion, *p* = 0.23).

Regarding the primary composite safety endpoint, ticagrelor exhibited similar bleeding risk, OR = 0.93, 95% CI, 0.82–1.05, *p* = 0.22 (Figure 3A), with low heterogeneity (I^2^ = 2%). Two studies reported data on major hemorrhagic events [16,18]. In the pooled analysis, ticagrelor therapy exhibited a similar risk of major bleeding as in patients receiving clopidogrel, OR = 0.83, 95% CI, 0.59–1.16, *p* = 0.27 (Figure 3B). Similarly, the risk of gastrointestinal bleeding did not reach statistical significance, OR = 0.81, 95% CI, 0.62–1.07, *p* = 0.13 (Figure 3C). However, the results are limited due to the small number of studies reporting the outcome.

Moreover, we performed a subgroup analysis in patients presenting with ACS, including acute myocardial infarction. The primary composite efficacy endpoint was reported in 3 studies [15,16,18] and was not statistically significant in the ticagrelor group as compared to clopidogrel, OR = 1.06, 95% CI, 0.78–1.44, *p* = 0.70 (Figure 4A), with high heterogeneity (I^2^ = 66%). Concerning the primary composite safety outcome, patients receiving ticagrelor displayed a similar risk as those from the clopidogrel group, OR = 0.96, 95% CI, 0.80–1.16, *p* = 0.68 (Figure 4B). All-cause mortality and cardiovascular mortality were similar in both treatment groups (Figure 4C,D), respectively OR = 1.16, 95% CI, 0.91–1.48, *p* = 0.24, and OR = 1.04, 95% CI, 0.47–2.30, *p* = 0.93. Ticagrelor had a similar efficacy profile regarding recurrent myocardial infarction and stroke occurrence (Figure 4E,F), respectively *p* = 0.73 and *p* = 0.21. Two studies investigated the impact of ticagrelor on major bleeding events in ACS patients [16,18]. The safety profile was similar to patients receiving clopidogrel (Figure 4G), *p* = 0.27.

The quality of included studies in the present meta-analysis was fair to good, as appraised by the NOS scale, despite the observational design of the studies (Appendix A). A funnel plot was used to display the publication bias (Figure 5).

## 4. Discussion

The optimal antiplatelet therapy for ESKD patients on chronic dialysis presenting with ACS or CCS remains uncertain due to limited data and conflicting guidelines. The PLATO trial, which demonstrated the superiority of ticagrelor over clopidogrel in reducing cardiovascular events, excluded patients requiring dialysis [7]. Similarly, the TWILIGHT study assessed the efficacy and safety of ticagrelor monotherapy and did not include patients on chronic dialysis [9]. These exclusions leave a significant research gap and hinder the ability to determine the optimal antiplatelet therapy for dialysis patients undergoing PCI.

The present meta-analysis aims to clarify these uncertainties by inquiring about available data on the comparative efficacy and safety of ticagrelor/clopidogrel in ESKD patients receiving chronic dialysis. Our findings highlight similar efficacy and safety of ticagrelor and clopidogrel, and the paucity of high-quality studies specifically addressing the use of more potent P2Y12 receptor inhibitors, such as ticagrelor, in dialysis patients. Despite the clinical relevance of this patient population, they have been systematically excluded from major trials comparing different antiplatelet regimens.

Studies have demonstrated that ticagrelor provides more potent and consistent platelet inhibition than clopidogrel (including patients with ACS), irrespective of clopidogrel resistance status [19]. Ticagrelor holds potential advantages due to its ability to overcome the limitations associated with clopidogrel resistance, thus reducing ischemic risk. Notably, clopidogrel resistance was reported in 25% of patients presenting with STEMI who underwent PCI, with a subsequent increased risk of adverse cardiovascular events. However, no data were mentioned on dialysis patients [20]. Therefore, ticagrelor proved to be superior to clopidogrel in clinical trials. In a recent meta-analysis, ticagrelor was linked to a reduced all-cause and cardiovascular mortality (OR = 0.68, 95% CI, 0.58–0.81 and OR = 0.64, 95% CI, 0.48–0.85, respectively), but with a higher incidence of major bleeding events (OR = 1.21, 95% CI, 1.06–1.39) [21]. Nevertheless, extrapolating these results to ESKD patients requiring dialysis might be misleading, as these patients exhibit different ischemic and hemorrhagic risk profiles.

The present meta-analysis included five observational studies that compared ticagrelor and clopidogrel in ESKD patients on chronic dialysis. The analysis showed no statistically significant differences in the primary composite efficacy outcome between the two antiplatelet drugs. The rates of all-cause mortality, cardiovascular mortality, coronary revascularization, recurrent myocardial infarction, and stroke were also similar between the two treatment groups. Also, the primary composite safety endpoint, which included major and non-major bleeding events, did not differ significantly between ticagrelor and clopidogrel. In subgroup analysis involving exclusively patients presenting with ACS, results were consistent, with a similar efficacy profile of ticagrelor and clopidogrel.

These findings suggest that ticagrelor might be as effective and safe as clopidogrel in ESKD patients on chronic dialysis in the context of both ACS and CCS. However, it is crucial to interpret the results with caution due to the limited number of studies and the observational nature of the included studies. Moreover, extensive studies with longer follow-up duration are required to investigate the long-term impact of ticagrelor and clopidogrel on adverse cardiovascular events. A more extended follow-up period is needed to evaluate the durability of treatment effects and the occurrence of delayed adverse events.

Three out of five studies included in the present meta-analysis were performed in Eastern Asia, where clopidogrel resistance was highly variable. Nevertheless, the efficacy and safety endpoints were similar in both treatment groups. In the sensitivity analysis (excluding studies from the USA), the results were consistent among dialysis patients treated with clopidogrel or ticagrelor. However, the results are limited due to the small number of patients, and large-scale randomized clinical trials are required to confirm these findings.

From the analyzed studies, Mavrakanas et al. investigated comparative efficacy and safety endpoints of ticagrelor and clopidogrel in dialysis patients who underwent PCI with drug-eluting stents (including for ACS) [17]. In this contemporary cohort of patients, results were concordant with the pooled effect from the current meta-analysis. The primary composite outcome (cardiovascular death, myocardial infarction, stroke) was not significantly different in both treatment groups (HR 1.0, 95% CI, 0.83–1.20). Also, no difference was reported for secondary outcomes and clinically relevant bleedings, supporting the overall effect of the meta-analysis [17].

The meta-analysis acknowledges the lack of well-designed randomized controlled trials or large-scale studies explicitly examining the safety and efficacy of ticagrelor and clopidogrel in dialysis patients. This limitation highlights the scarcity of robust evidence in this subset of high-risk patients. Heterogeneity across studies was high, which might impact the overall reliability and generalizability of the pooled results. Nevertheless, the effect was similar in sensitivity analysis, following sequentially excluding investigated studies.

Despite these limitations, the present meta-analysis provides valuable insights into the efficacy and safety of ticagrelor and clopidogrel in patients with ESKD and dialysis, highlighting the need for further research in this area.

## 5. Conclusions

The meta-analysis’s evidence suggests no significant difference in the efficacy and safety outcomes between ticagrelor and clopidogrel in ESKD patients on chronic dialysis who require antiplatelet therapy for ACS or CCS. Although there was a trend towards a lower incidence of adverse cardiovascular events with ticagrelor compared to clopidogrel, the difference did not reach statistical significance. Also, the risk of major bleeding events was similar between ticagrelor and clopidogrel in this subset of patients. However, the available evidence is limited, and more well-designed randomized controlled trials or large-scale studies are needed to assess the comparative efficacy and safety of ticagrelor and clopidogrel in dialysis patients. The findings of this meta-analysis also highlight the requirement for further research specifically focusing on dialysis patients to guide the optimal antiplatelet therapy in this population and improve treatment outcomes.

## Figures and Tables

**Figure 1 jcm-12-05011-f001:**
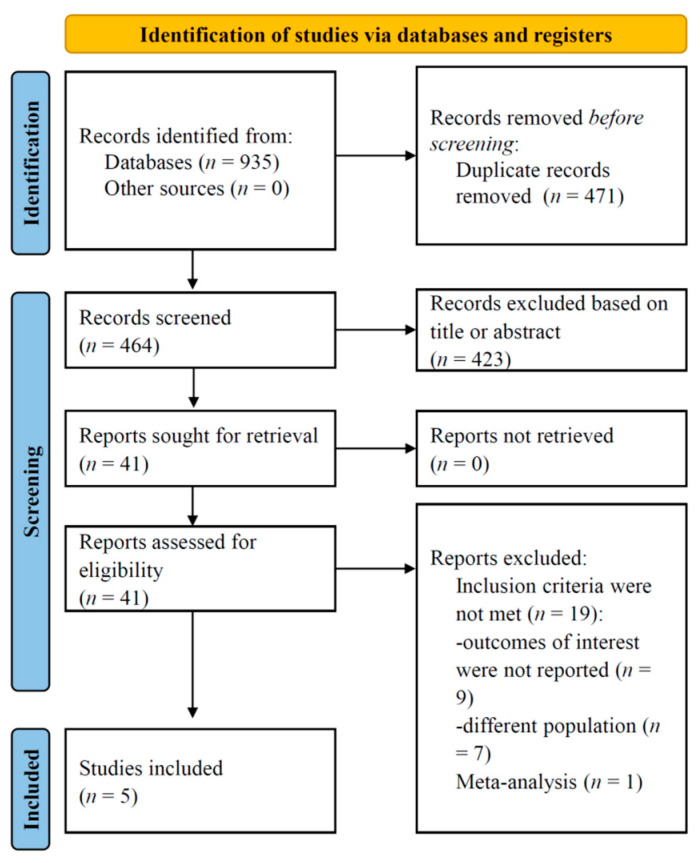
PRISMA flow-diagram of the search process.

**Figure 2 jcm-12-05011-f002:**
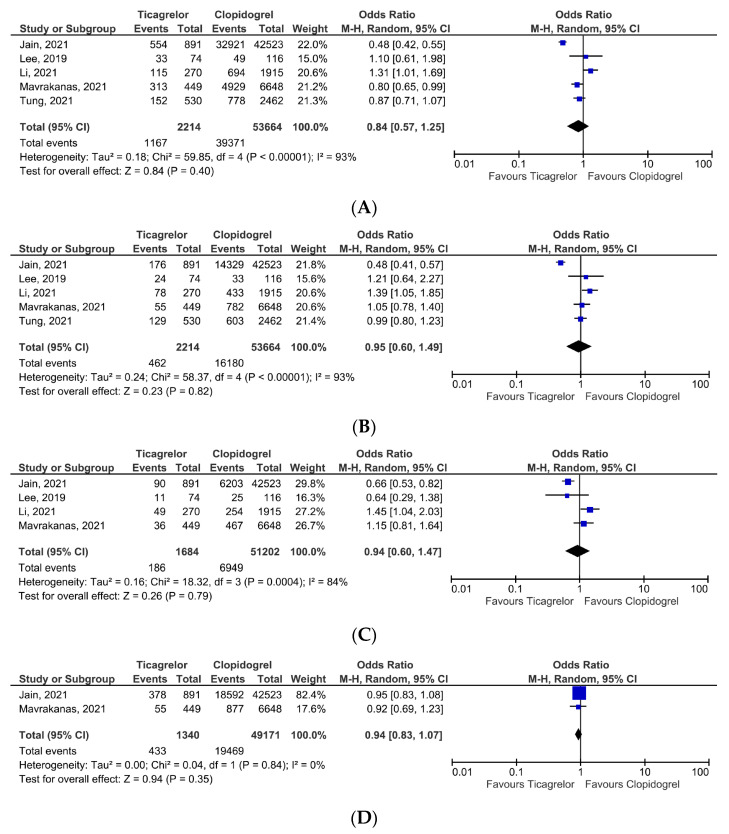
(**A**) Primary composite outcome-efficacy. (**B**) All-cause mortality. (**C**) Cardiovascular mortality. (**D**) Coronary revascularization. (**E**) Recurrent myocardial infarction. (**F**) Ischemic or hemorrhagic stroke [14,15,16,17,18].

**Figure 3 jcm-12-05011-f003:**
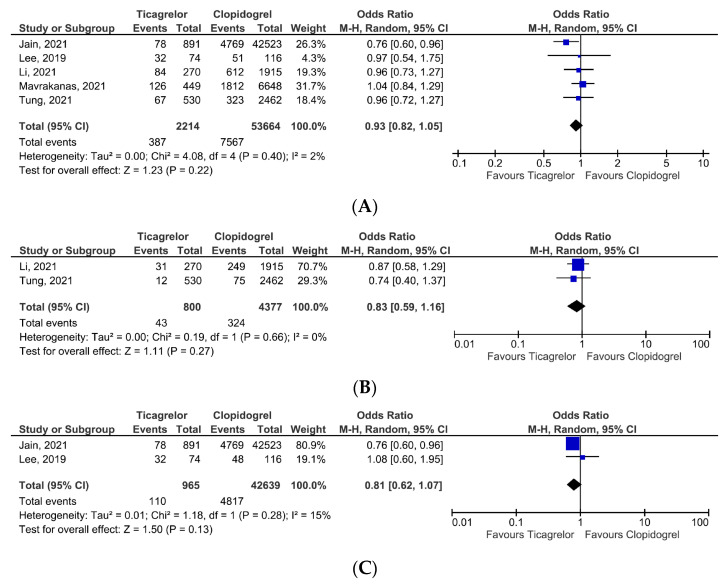
(**A**) Primary composite outcome-safety. (**B**) Major bleeding events. (**C**) Gastrointestinal bleeding events [14,15,16,17,18].

**Figure 4 jcm-12-05011-f004:**
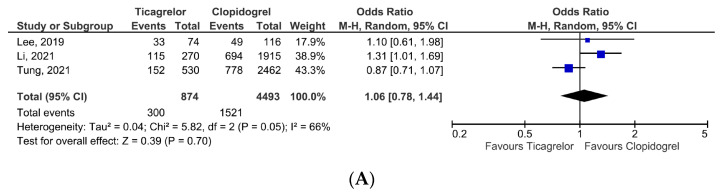
(**A**) Primary composite outcome-efficacy (ACS patients). (**B**) Primary composite outcome-safety (ACS patients). (**C**) All-cause mortality (ACS patients). (**D**) Cardiovascular mortality (ACS patients). (**E**) Recurrent myocardial infarction (ACS patients). (**F**) Ischemic or hemorrhagic stroke (ACS patients). (**G**) Major bleeding events (ACS patients) [15,16,18].

**Figure 5 jcm-12-05011-f005:**
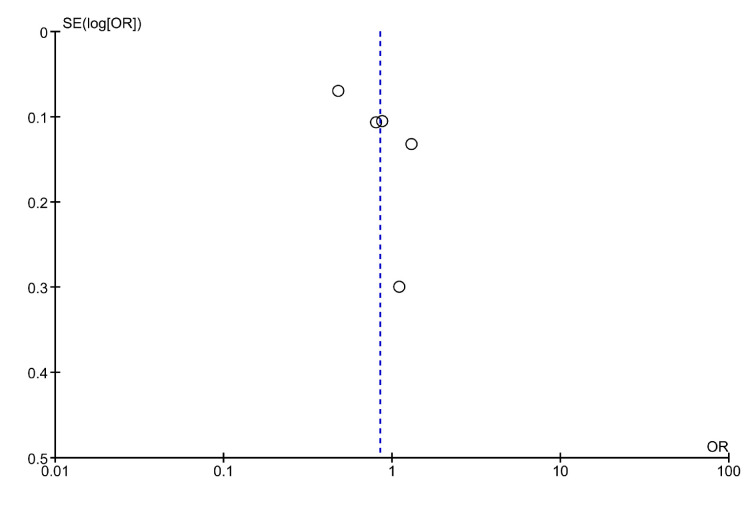
Funnel plot of included studies.

**Table 1 jcm-12-05011-t001:** General characteristics of studies included in the present systematic review and meta-analysis.

Author, Year	Design	Patients, No	Age, Median/Mean ± SD	Setting	Outcomes	Follow-Up Period
Jain et al., 2021 [14]	Observational, retrospective (USA)	42,523 (clopidogrel group)	64.0	Patients with ESKD who were prescribed P2Y12 receptor inhibitors.	The primary outcome: all-cause death. The secondary outcomes: cardiovascular death, coronary revascularization and gastrointestinal bleedings.	Median follow-up 52 weeks.
		891 (ticagrelor group)				
Lee et al., 2019 [15]	Observational, retrospective (Taiwan)	116 (clopidogrel group)	67.90 ± 9.18	Chronic dialysis patient who underwent PCI for acute myocardial infarction (both, STEMI and NSTEMI).	The primary composite outcome: cardiovascular death, recurrence of myocardial infarction, or new stroke. The secondary outcomes: cardiovascular death, all-cause death, recurrence of myocardial infarction, new stroke and any bleeding event.	In-hospital and at 1-year.
		74 (ticagrelor group)	65.19 ± 10.42			
Li et al., 2021 [16]	Observational, retrospective (Taiwan)	1915 (clopidogrel group)	67.20 ± 11.26	Patients receiving chronic dialysis who presented with acute coronary syndrome.	The primary efficacy outcome: MACE occurrence (composite of any-cause mortality, recurrent myocardial infarction and stroke). The primary safety outcome: major bleeding events (requiring hospitalization or admission to emergency room). The secondary outcomes: all-cause death, cardiovascular death, recurrent myocardial infarction, stroke, and any bleeding events.	Until outcome occurrence, death, 12 months or the end of the study.
		270 (ticagrelor group)	64.24 ± 11.65			
Mavrakanas et al., 2021 [17]	Observational, retrospective (USA)	6648 (clopidogrel group)	64 ± 11	Renal replacement therapy patients (hemodialysis or peritoneal dialysis) who underwent PCI with drug-eluting stent implantation (including for acute coronary syndrome).	The primary composite outcome: cardiovascular mortality, myocardial infarction or stroke. The secondary outcomes: individual components of the primary outcome; composite of cardiovascular mortality, myocardial infarction, stroke or coronary revascularization; all-cause mortality; clinically relevant bleedings or any bleeding requiring hospitalization.	Until death, kidney transplant, loss of coverage or 12 months after stent implantation.
		449 (ticagrelor)	64 ± 12			
Tung et al., 2021 [18]	Observational, retrospective (Taiwan)	2462 (clopidogrel group)	29.9% patients > 75 years	Patients on chronic hemodialysis admitted with acute myocardial infarction.	The primary composite outcome: all-cause mortality, non-fatal myocardial infarction, or non-fatal stroke. The secondary outcomes: individual components of the primary composite outcome. The safety outcome: BARC type 2, 3, or 5 bleedings.	9 months, or until outcomes occurrence.
		530 (ticagrelor group)	24.9% patients > 75 years			

BARC = Bleeding Academic Research Consortium; ESKD = end-stage kidney disease; NSTEMI = non-ST-elevation myocardial infarction; PCI = percutaneous coronary intervention; STEMI = ST-elevation myocardial infarction.

**Table 2 jcm-12-05011-t002:** Results reported in studies included in the present systematic review and meta-analysis.

Author, Year	Outcomes	Results (Ticagrelor vs. Clopidogrel)	
Jain et al., 2021 [14]	All-cause death	In propensity matched cohorts: HR 1.05 (95% CI, 0.90–1.24)	
	Cardiovascular death	In propensity matched cohorts: HR 1.08 (95% CI, 0.86–1.35)	
	Coronary revascularization	In propensity matched cohorts: HR 0.99 (95% CI, 0.92–1.06)	
	GI bleeding	In propensity matched cohorts: HR 0.82 (95% CI, 0.65–1.03)	
Lee et al., 2019 [15]	The primary composite outcome	The primary composite outcome had similar incidence at 1-year in both treatment groups (free of composite outcome 72.16% vs. 66.06%)	*p* = 0.424
	Cardiovascular death	No statistically significant differences were reported (free of cardiovascular death 83.62% vs. 72.20%)	*p* = 0.372
	Any-cause death	All-cause death was similar in both treatment arms (free of death 70.24% vs. 64.85%)	*p* = 0.446
	Recurrent MI	Free of myocardial infarction was similar in both groups (85.47% vs. 81.98%)	*p* = 0.406
	Stroke	Patients receiving ticagrelor had no stroke reported, while 4 patients from clopidogrel group experienced stroke	*p* = 0.117
	Bleeding event	Bleeding incidence was similar in both groups (free of bleeding 56.53% vs. 54.42%)	*p* = 0.664
Li et al., 2021 [16]	MACE	HR 1.29 (95% CI, 1.16–1.44)	*p* < 0.0001
	Any-cause death	HR 1.65 (95% CI, 1.47–1.86)	*p* < 0.0001
	Cardiovascular death	HR 1.64 (95% CI, 1.41–1.91)	*p* < 0.0001
	Recurrent MI	HR 0.93 (95% CI, 0.75–1.16)	*p* = 0.5063
	Stroke	HR 0.94 (95% CI, 0.75–1.19)	*p* = 0.6292
	Major bleedings	HR 1.49 (95% CI, 1.34–1.65)	*p* < 0.0001
	Any bleedings	HR 1.05 (95% CI, 0.95–1.17)	*p* = 0.3506
Mavrakanas et al., 2021 [17]	The primary composite outcome	HR 1.00 (95% CI, 0.83–1.20)—main analysis	
	Cardiovascular death	HR 1.17 (95% CI, 0.75–1.82)—main analysis	
	MI	HR 1.04 (95% CI, 0.83–1.31)—main analysis	
	Stroke	HR 1.04 (95% CI, 0.82–1.32)—main analysis	
	Coronary revascularization	HR 1.19 (95% CI, 0.88–1.62)—main analysis	
	Clinically relevant bleeding	HR 1.13 (95% CI, 0.91–1.40)—main analysis	
Tung et al., 2021 [18]	The primary composite outcome	HR 1.16 (95% CI, 0.97–1.39)	*p* = 0.11
	All-cause death	HR 1.17 (95% CI, 0.97–1.42)	*p* = 0.11
	Non-fatal MI	HR 1.05 (95% CI, 0.66–1.66)	*p* = 0.84
	Any bleeding	HR 1.25 (95% CI, 0.96–1.63)	*p* = 0.09
	BARC type 3 or 5 bleeding	HR 0.93 (95% CI, 0.51–1.70)	*p* = 0.82

BARC = Bleeding Academic Research Consortium; GI = gastrointestinal; MACE = major adverse cardiovascular event; MI = myocardial infarction.

## Data Availability

Not applicable.

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
