# Peer review of "Efficacy and Safety of Ticagrelor versus Clopidogrel in Dialysis Patients with Coronary Syndromes: A Systematic Review and Meta-Analysis"

_jcm, 2023, doi:10.3390/jcm12155011_

Round 1

Reviewer 1 Report

The manuscript by Burlacu and coworkers reported a systematic review and the meta-analysis to compare the efficacy and safety of ticagrelor and clopidogrel in dialysis patients presenting with coronary syndromes. The head-to-head comparison revealed no difference in the efficacy and safety between ticagrelor and clopidogrel; the primary composite efficacy outcome included a combination of all-cause and cardiovascular mortality, etc and the primary safety outcome included major and non-major bleeding events. This is a significant observation as this patient group were absent in major clinical trials supporting approval of ticagrelor and clopidogrel. The results have clinical implications to guide use of dual antiplatelet therapy. I recommend  this well-written manuscript after the authors address the following:

The authors based their conclusion on five clinical trial data (Table 1). It would be helpful if the authors provided data of ethnicity.  It is well known that response to clopidogrel is highly variable among patients of different ethnic groups due to CYP2C19 loss-of-function polymorphism, whereas response to ticagrelor is not. Would the authors reach the same conclusion if the ethnic background of dialysis patients were considered?

A few typos need to be corrected:

Line 93, delete “The” at the start of the sentence.

Line 154, delete “This” at the start of the sentence.

Reviewer 2 Report

The chief limitation is small numbers of patients and non-randomized studies used in this meta-analysis.  However the authors acknowledge these limitations.  Their interpretation is appropriately conservative.  Altogether this is a carefully executed and well-written study.

This is the one paper in 50 that I review where I do not find any criticisms or comments that would be helpful
